# Energy Potential of Urban Green Waste and the Possibility of Its Pelletization

Vladimír Mašán [1,*], Patrik Burg [1], Jiří Souček [2], Vlastimil Slaný [3] and Lukáš Vaštík [1]

[1] Department of Horticultural Machinery, Faculty of Horticulture, Mendel University in Brno, Valtická 337, 691 44 Lednice, Czech Republic; patrik.burg@mendelu.cz (P.B.); xvastik@node.mendelu.cz (L.V.)

[2] Research Institute of Agriculture Engineering, p. r. i., 161 01 Prague, Czech Republic; jiri.soucek@vuzt.cz

[3] Department of Agricultural, Food and Environmental Engineering, Faculty of AgriSciences, Mendel University in Brno, Zemědělská 1665/1, 613 00 Brno, Czech Republic; vlastimil.slany@mendelu.cz

* Correspondence: vladimir.masan@mendelu.cz

**Abstract:** Due to ongoing changes in the European energy market, there is currently a need to find new and additional uses for waste materials. Horticultural waste, which has not yet been fully recognized, offers a relatively wide potential in this area. Although the properties of these wastes are not ideal for combustion, they can be used as a solid biofuel. The pellets that are produced, however, may have useful properties, either energetic or environmental, and are valuable when utilized in boilers. In this study, six examples of typical input raw materials were selected, analysed, and subsequently pelletized. The experimental results provided an overview of the physio-chemical properties of the evaluated samples. Specifically, the moisture content (9.2–27.8%), bulk density (131.4–242.8 kg·m$^{-3}$ wt), ash content (3.0–28.0%), lower calorific value (11.3–16.2 MJ·kg$^{-1}$), and major and minor elements, were evaluated. The pelletization process and resultant pellet characteristics, such as durability (96.3–98.8%), moisture content (7.5–11.5%), and dimensions, were also evaluated. In the statistical evaluation, significant differences were found between individual samples. In particular, both the branches and the mixture of perennial plants met the industry standard limits, showing that they are of sufficient quality. On the contrary, the sample of fallen leaves was particularly problematic with regard to a number of parameters (moisture, ash content, and calorific value). The overview of the analyses performed expands the current state of knowledge on the potential to use selected types of horticultural waste in the field of energy and for the production of shaped biofuels.

**Keywords:** bioenergy; sustainability; solid biofuels; pellets; high heating value; chemical composition

## 1. Introduction

In recent years, for generally well-known reasons, significant changes have occurred in all EU member states in terms of energy production and consumption. Manufacturing within most EU countries is dependent on the import of energy or raw materials. More than half of the energy consumed by EU member states is imported, with the largest share coming from oil, natural gas, and coal [1]. As such, the EU is trying to direct its energy sector towards greener sources, which, in addition to reducing its dependence on imports, has the potential to solve a number of environmental problems and position the EU as a leader in green energy. The goals of the Green Deal and other EU strategies are directed towards meeting this vision [2].

Directive (EU) 2018/2001 of the European Parliament obliges member countries to increase the share of renewable energy sources in their total energy consumption by at least 40% by 2030 [3]. At the same time, the Fit for 55 plan establishes a need to reduce net emissions by more than 50% compared to 1990, making Europe a climate-neutral continent [4]. Moreover, the European Commission proposed the REPowerEU strategic

plan, the main objective of which is to strengthen energy independence and halt imports from Russia [5].

Renewable energy from waste biomass is not a new energy source; it has long been in use, but potential new options and sources that allow its use are still under exploration [6]. Waste biomass from the maintenance of urban green areas has great potential to solve many of these problems, but it is still only partially used [7].

Waste from municipal maintenance can account for up to 10–20 percent of the overall amount of municipal waste. This waste is available throughout the year at almost the same quality and moisture levels, particularly solid and dry waste (pruned wood from trees and shrubs, dry plant matter, fallen leaves, etc.) from public green spaces and gardens [8–10].

This waste can be used directly in cities as a local energy source and has the potential for use in producing shaped biopellets. Solid biofuels are popular from both the perspective of the technology, which is simple and has already been adopted by many companies, and their ease of use in terms of the price and widespread use of boilers [11–13].

The pellet production process is less harmful to the environment (with energy being used instead of chemicals) than the bioethanol production process [14].

Since 2004, there has been a 10% per annum increase in the consumption of wood pellets in the EU member states. The total consumption reached $23.1 \times 10^6$ tons of wood pellets in 2021 [15]. The growing demand for pellets has contributed to the search for non-wood raw materials to ensure the availability of this type of biofuel. The transition to non-wood pellets has also been fuelled by the growing fear that, over the long term an unsustainable quantity of wood is being harvested.

Non-wood raw materials can act as binders for less suitable forms of biomass to improve the durability and physical quality of pellets, reduce the amount of dust, improve pelleting efficiency, and reduce energy costs [16–19]. However, they can also have a lower calorific value, higher ash content, and higher emissions, and, in extreme cases, cause corrosion and sintering during the combustion process [20–28]. First, we need to determine the energetic characteristics of each material as potential options for modification.

The aim of this study is to analyse the energy potential and selected physio-chemical properties of waste products generated during the maintenance of parks and communal greenery, along with an evaluation of their suitability for pellet production.

## 2. Materials and Methods

The waste materials (Figure 1) were collected from the public green and ornamental flower beds at the Faculty of Horticulture, Mendel University, Lednice, CZ (48.795601 N, 16.798340 E). The green areas were treated the same as they would be if they were subjected to the normal maintenance of public green areas. Therefore, these results are highly representative and usable in practice.

### 2.1. Climatic Conditions

The region is characterized by significantly warm weather with an average annual temperature of 9 °C, relative humidity of 80%, and long-term annual rainfall of 500 mm. The average relative humidity is approximately 80%. The weather patterns for 2022 can be found on the website [29].

### 2.2. Samples

Materials that had not yet been subjected to verification in a similar context were selected and were in minimal common practical use. The materials were, however, produced in relatively large quantities during the maintenance of urban green areas. Table 1 provides an overview of the basic characteristics of the raw samples.

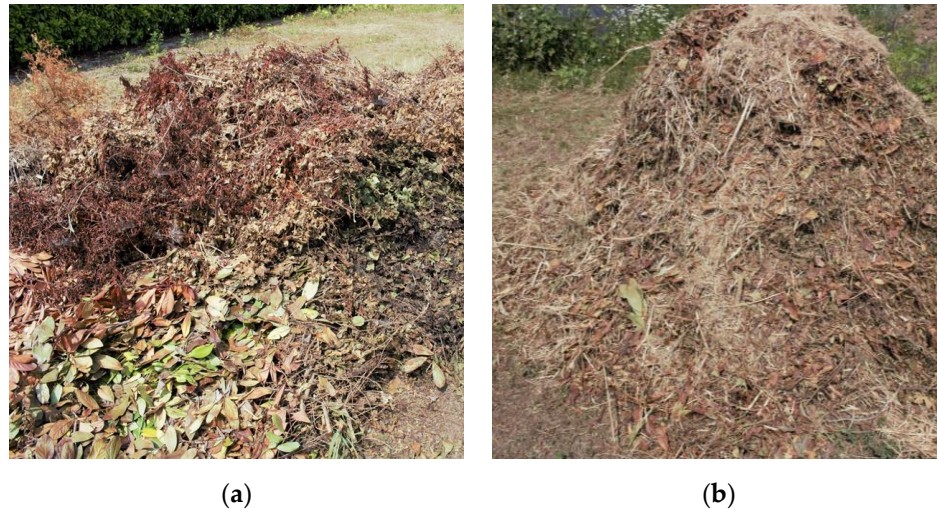

(**a**)                                (**b**)

**Figure 1.** Material of sample M1 used in this study: (**a**) raw material; (**b**) crushed material.

**Table 1.** Basic characteristics of the raw samples.

| ID | LW | BW | BB | BC | M1 | M2 |
|---|---|---|---|---|---|---|
| Sample/ Parameter (Unit) | Walnut Leaves | Walnut Branches | Black Locust Branches | Common Apricot Branches | Mixture 1 Branches | Mixture 2 Perennials |
| Moisture at collection (%) | 27.83 ± 1.75 [d] | 10.86 ± 0.26 [a] | 13.47 ± 0.09 [b] | 16.06 ± 0.22 [c] | 9.21 ± 0.06 [a] | 10.00 ± 0.39 [a] |
| HHV (MJ·kg$^{-1}$) * moisture at 10% | 12.49 ± 0.74 [c] | 17.67 ± 0.11 [a] | 17.12 ± 0.09 [ab] | 16.35 ± 0.08 [b] | 17.4 ± 0.07 [a] | 17.09 ± 0.16 [ab] |
| LHV (MJ·kg$^{-1}$) | 11.32 ± 0.69 [c] | 16.15 ± 0.09 [a] | 15.61 ± 0.07 [ab] | 14.85 ± 0.06 [b] | 15.97 ± 0.05 [a] | 15.60 ± 0.13 [ab] |
| Bulk density (kg·m$^{-3}$ wt) | 140.31 ± 11.78 [a] | 242.82 ± 3.14 [d] | 173.97 ± 3.68 [c] | 196.61 ± 7.22 [b] | 201.57 ± 9.93 [b] | 131.45 ± 4.46 [a] |
| Ash (wt%) | 28.05 ± 1.32 [c] | 5.41 ± 2.40 [ab] | 3.02 ± 0.23 [a] | 7.2 ± 1.79 [ab] | 5.96 ± 1.49 [ab] | 9.49 ± 1.33 [b] |
| C (wt%, dry matter) | 37.49 | 49.02 | 48.01 | 47.73 | 46.81 | 47.98 |
| H (wt%, dry matter) | 4.23 | 5.88 | 5.79 | 5.73 | 5.43 | 5.67 |
| O (wt%, dry matter) | 28.56 | 38.55 | 41.76 | 37.40 | 40.69 | 35.62 |
| N (wt%, dry matter) | 1.34 | 0.94 | 1.21 | 1.72 | 0.80 | 0.91 |
| O:C | 0.76 | 0.79 | 0.87 | 0.78 | 0.87 | 0.74 |

Note: Data are expressed as means ± standard deviation. Means with different letters within the row are significantly different, according to Tukey's test ($p \leq 0.05$). * The samples were dried in a climate box to a uniform moisture of 10%.

The chosen material were branches from Walnut (*Juglans regia* L.)—BW, Black Locust (*Robinia pseudoacacia* L.)—BB, and Common Apricot (*Prunus armeniaca* L.)—BC, along with fallen leaves from Walnut plants (*Juglans regia* L.). Mixture 1 (M1) was made up of 50% branches of woody plants and 50% ornamental grass. Mixture 2 (M2) was made up of the above-ground parts of ornamental perennials (with a predominance of Mediterranean species, such as *Lavender* sp., *Thyme* sp., and *Catmint* sp.).

These specific materials and mixtures were waste materials that are commonly generated by the maintenance of municipal green areas and so far have only been used for energy purposes to a limited extent. In particular, walnut and apricot branches are good examples of materials that come from pruning popular urban extensive orchards. Agate branches represent so-called avenue trees, the maintenance of which is rather short-term. Walnut leaves are unsuitable for composting and are a good example of autumn waste that is wet and possibly contaminated. M1 is a material mix that comes from the summer maintenance of greenery when the individual components of fleshy shoots, wood, and leaves are not separated. M2 is a mix that comes from the maintenance of ornamental beds of grasses, annuals, and perennials, which mainly takes place during autumn.

The materials were separately harvested for use as individual variants in the experiments (approximately 300 kg per variant). The collected samples were immediately crushed

using an SPK 30 cutting mill (Kovo Novák, Czech Republic) with 5 mm sieves for analysis and pelletization [30]. The samples for analysis were crushed using a Retsch SM 100 cutting mill (Retsch GmbH, Haan, Germany) using the required sieve fractions according to the relevant standards (most often 0.5 mm) (Figure 2). The chips were analysed in a biofuel laboratory to determine the moisture content in their fresh state after collection, followed by drying in a climate box to a moisture content of 10%. The main physio-chemical parameters were subsequently determined for the samples prepared in this way.

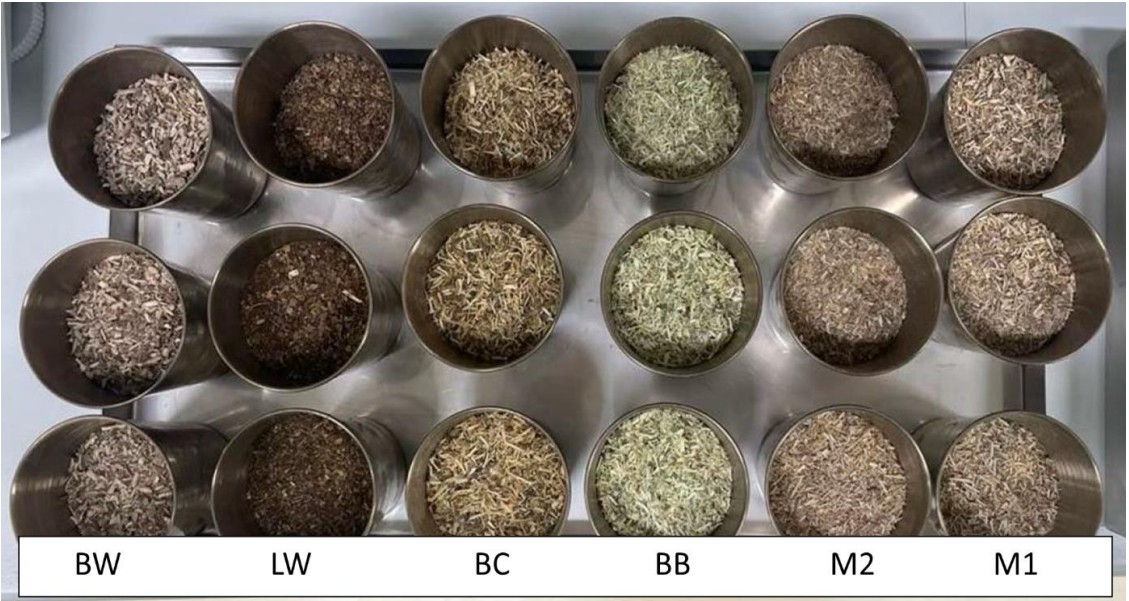

**Figure 2.** Crushed samples prepared for analysis on sieves with aperture sizes of 0.5 mm. BW—Branches from Walnut (*Juglans regia* L.); BB—Black Locust (*Robinia pseudoacacia* L.); BC—Common Apricot (*Prunus armeniaca* L.); LW—fallen leaves from Walnut (*Juglans regia* L.); M1—mixture of 50% woody plants and 50% ornamental grasses; M2—mixture of above-ground parts of ornamental perennials.

### 2.3. Physical and Chemical Properties

The chemical and physical analyses were conducted as per the relevant ISO standards. The samples for analysis were weighed using an Ohaus PX224 laboratory scale (OHAUS Europe GmbH, Nänikon, Switzerland). The bulk density of the samples was measured according to the ISO 17828 standard [31]. The moisture content of the samples was measured according to the method described in ISO 18134-2 [32]. The samples were placed in an oven at 105 ± 2 °C until a constant weight was attained. The ash content was measured according to ISO 18122 [33] in a LAC LMH muffle furnace (LAC, s.r.o., Židlochovice, Czech Republic). The samples of the chips (minimum 1 g each) were weighed before and after their complete combustion at 550 ± 10 °C. The higher heating value (HHV) of the samples was analysed using a Parr 6400 automatic isoperibol calorimeter (Parr Instruments, Moline, IL, USA) according to ISO 18125 [34]. The lower heating value (LHV) was calculated from the HHV in accordance with the equations in ISO 1928 [35]:

$$Q_i^r = Q_s^r - \gamma \cdot (W_t^r + 8.94 \cdot H_t^r) \tag{1}$$

where

- $Q_i^r$—LHV of the evaluated sample, MJ·kg$^{-1}$;
- $Q_s^r$—HHV of the original sample, MJ·kg$^{-1}$;
- $\gamma$—ratio of evaporation of 1% $H_2O$, MJ·kg$^{-1}$, at temp. 25 °C, $\gamma$ = 0.02442 MJ·kg$^{-1}$;
- 8.94—hydrogen to water conversion ratio of, –;
- $W_t^r$—total water content in the original sample, %;
- $H_t^r$—total hydrogen content in the original sample, %.

### 2.4. Determination of Carbon, Hydrogen and Nitrogen

The amounts of the major elements (C, H, and N) were determined using a combustion method, the LECO CHN628 + S instrumental LECO combustion method for biomass (LECO Corporation, Saint Joseph, MI, USA), according to ISO 16948 [36]. The standards used in the calibration were taken from LECO: EDTA (ethylenediaminetetraacetic acid) and the results were automatically calculated.

### 2.5. Determination of the Content of Major and Minor Elements, Total Sulphur, and Total Chlorine

The chemical components were determined using X-ray fluorescence. An analysis was performed using a Niton XL3t GOLDD+ (Thermo Fisher Scientific, Waltham, MA, USA), with static and cuvettes based on ISO 16967 [37], ISO 16968 [38], and ISO 16994 [39]. The AllGeo method was used. The accuracy of the analyses was checked using the reference materials (IRM 5718, Metranal 6, Metranal 13, Metranal 22, NIST 2781, and NIST 2702). The results were expressed as % and then calculated as $mg \cdot kg^{-1}$.

### 2.6. Pellet Production

The raw materials were pelletized using a BONSAI 100 pelletizer (Kovo Novák, Citonice, Czech Republic) with a plate matrix equipped with 6 mm diameter holes and 25 mm in length (Figure 3). The pelletizer automatically ejected the pellets onto a rotary sieve screen with a cooler. The pressing pressure was adapted to the specific material, with a maximum output of $120 \ kg \cdot h^{-1}$. The press was driven by a 6 kW electric motor. According to general recommendations [18,40,41], a moisture level of less than 15% was chosen for the production of pellets, which was adjusted during pelleting to achieve the strongest pellets.

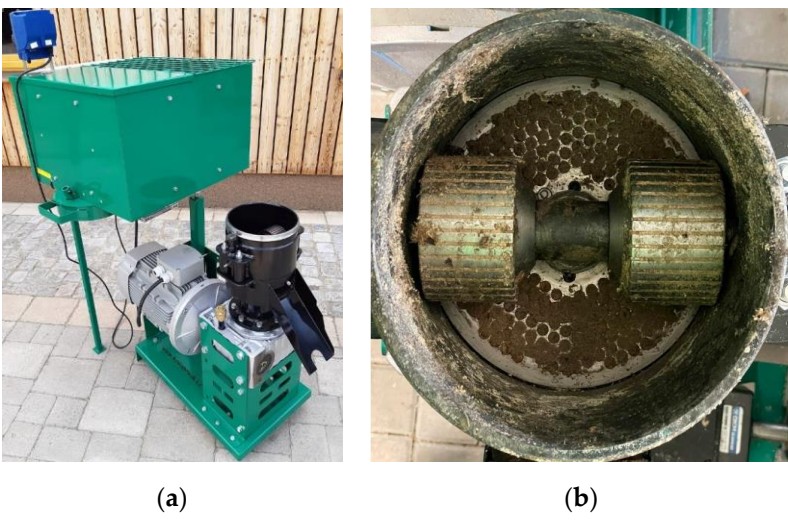

(**a**)　　　　　　　　　　　　　　(**b**)

**Figure 3.** (**a**) BONSAI 100 pelletizer; (**b**) extrusion rollers and die plate with holes.

### 2.7. Evaluated of the Pellets Parameters

The pellet dimensions were determined using a digital calliper gauge Dasqua IP67 (Dasqua S.R.L., Cornegliano Laudense, Italy). The measurements were performed 20 times for each sample. The mechanical durability was determined using a wear resistance analysis device, PT 500 (Research Institute of Agricultural Engineering p.r.i., Prague, Czech Republic), as per ISO 17831-1 for pellets [42].

### 2.8. Methods of Statistical Analysis

The data obtained were analysed using available statistical methods with MS Excel and Statistica 14.0 software packages (StatSoft Inc., Tulsa, OK, USA). The results were expressed as means and standard deviations. An analysis of variance and Tukey's post hoc tests, with a significance level of α = 0.05, were used for detailed comparisons.

## 3. Results and Discussion

### 3.1. Biomass Potential

The moisture content, chemical composition, bulk mass, energy content (HHV and LHV), and ash content of the biomass samples at collection are listed in Table 1. These are the most important parameters from the perspective of the suitability of the selected biomass for utilization as an energy source, biofuel production, and meeting emission limits [43]. Biomass that is similar to wood (BW, BC, and BB) or herbaceous parts (LW, M1, and M2) shows similar characteristics [44].

#### 3.1.1. Moisture

According to the results, the highest moisture content was found in LW and the lowest in the M1 and M2 samples. This corresponds with both the nature of the material and the time of collection. Sample LW was collected during the wet autumn; M1 and M2 were collected during the summer. Furthermore, the wooden materials, BB, CB, and WB, had similar moisture contents, which were the lowest levels measured, as expected.

Moisture during collection significantly affected the ability to handle and store the material, the requirement to dry it, and thus the overall efficiency of the biofuel production process [10]. However, the minimum moisture content of the material, estimated at 10–20%, was advantageous for good pelletization of the material [40,41]. In general, the moisture content of the samples is reduced when pelletized.

Pradhan et al. [45] reported a moisture range of 5 to 35% for garden waste; for comparison, sawdust has a moisture content from 11 to 41% [46].

#### 3.1.2. Bulk Density

None of the samples exceeded the upper limit of bulk density for dried wood chips of 260 kg·m$^{-3}$; rather, they all remained close to the lower limit with an average of 204 kg·m$^{-3}$ [47]. The results for LW and M2 did not even reach the values for soft wood, whose dry bulk density was in the range of 150–165 kg·m$^{-3}$. The measured values of bulk density depended on the type and composition of the material (herb and leaf components or wood) [48,49].

#### 3.1.3. Ash Content

Ash content is an expression of the biomass quality. A high ash content can cause higher levels of pollutant emissions, fouling, slagging, corrosion, and deposit formation problems in boilers, ultimately reducing the heating value [28,46,50,51].

The ash content of the wood waste materials (BW, BB, BC, and M1) was, on average, lower than that of the green waste materials (LW and M2). Furthermore, the energy potential of these raw materials was also higher. These results were expected, but the differences, with the exception of the BC ash content and the energy potential of LW, were not significant.

Tanoh et al. [44] state that the ash content for green waste pellets in their results was as high as 2.69 wt%, and for wood pellets, it was 0.88 wt%. Nasrin et al. [52] determined an ash content greater than 6%.

The upper limits for the ash content of materials used to heat family homes are ≤0.7% (class A1), ≤1.2% (class A2), and ≤2% (class B) [53]. The limits for industrial applications are more relaxed: ≤1% (for I1), ≤1.5% (for I2), and ≤3% (for I3). Therefore, the analysed samples did not meet the low ash content requirement and thus cannot be used alone as a raw material for the production of pellets.

Similarly, according to ISO 17225-6, graded non-woody pellets [54], such as the materials in LW, also failed to meet the limits of ≤6% (class A) and ≤10% (class B) because of their high ash content. Only the M2 sample met the class B limit.

Many authors would agree that this is a consequence of the high levels of bark in the branch samples and straw-like materials, which have a lower lignin content in the LW and M2 samples [55–57]. On the other hand, these materials could be used in mixtures to

reduce the amount of woody material used and improve other properties in the pelletizing process (how well the materials stick together, the durability of the pellets, etc.).

### 3.1.4. Higher Heating Value

The most important characteristic of fuel is its heating value, which is the amount of heat per kg. The analysis of the heating value of the raw materials showed values from 14.85 to 16.15 MJ·kg$^{-1}$. LW only produced 11.31 MJ·kg$^{-1}$, which did not meet the minimum requirement of ISO 17225-6 [54] and stipulated a minimum value of 14.5 MJ·kg$^{-1}$. This could be due to the lower carbon or lignin content [55–57]. On the other hand, it is notable that M2 did not have a significantly lower LHV than the more wooden materials. This could be due to the higher content of essential oils found in perennial herbaceous plants, which increased their calorific value. The wood samples had a lower LHV than the woody materials used in other studies, with an average of 18.5 MJ·kg$^{-1}$. This could be due to the higher proportion of bark and less mature wood in the branches [6,58].

According to Tukey's HSD test, significant differences were revealed between the HHVs for LW, BC, and other samples, and also between the LW and other samples for ash content.

The high carbon and hydrogen contents and low oxygen content were responsible for the high heating values [16]. Fuels with low ratios of O:C and H:C were considered to be appropriate for use in combustion. They produced a low amount of water vapour and smoke and lost a minimum amount of energy during combustion [59]. The organic compositions of the samples were highly similar, except for those of LW. This was also the reason why all samples had similar energy values. Of the selected materials, only the fallen leaves were unsuitable for direct burning.

### 3.1.5. Chemical Composition

For the natural waste materials and biomass, there are usually no problems with excessive levels of the monitored elements. This was confirmed by the current samples when none of the limits set in the standards (N, S, Cl, As, Cd, Cr, Cu, Pb, Hg, Ni, and Zn) were exceeded (Table 2).

**Table 2.** Chemical composition of samples (minor elements).

| ID | LW | BW | BB | BC | M1 | M2 |
|---|---|---|---|---|---|---|
| Sample /Elements (mg·kg$^{-1}$) | Walnut Leaves | Walnut Branches | Black Locust Branches | Common Apricot Branches | Mixture 1 Branches | Mixture 2 Perennials |
| Cd | 10.61 ± 0.92 | 12.39 ± 3.27 | 17.16 ± 2.25 | 16.96 ± 2.51 | 13.06 ± 1.63 | 12.23 ± 2.50 |
| Pd | nd | 5.33 ± 0.55 | nd | nd | 4.77 ± 0.57 | 5.18 ± 0.82 |
| Mo | 4.23 ± 0.22 | 5.53 ± 0.75 | 7.12 ± 0.05 | 6.28 ± 0.77 | 6.03 ± 0.30 | 6.41 ± 0.43 |
| Nb | 6.93 ± 0.36 | 6.6 ± 0.55 | 8.07 ± 1.19 | 8.87 ± 0.34 | 7.45 ± 0.96 | 7.5 ± 1.48 |
| Zr | 34.76 ± 0.85 | 5.47 ± 0.47 | 7.06 ± 1.69 | 6.68 ± 0.83 | 8.59 ± 0.51 | 15.87 ± 0.70 |
| Sr | 106.53 ± 1.05 | 45.48 ± 0.19 | 64.74 ± 1.42 | 47.58 ± 1.30 | 29.7 ± 0.81 | 35.64 ± 0.49 |
| Rb | 8.7 ± 0.29 | 2.84 ± 0.17 | 2.35 ± 0.47 | 7.57 ± 0.09 | 4.41 ± 0.37 | 2.93 ± 0.13 |
| Ti | 873.47 ± 21.37 | 44.92 ± 1.09 | 84.12 ± 3.55 | 84.4 ± 4.95 | 95.51 ± 16.89 | 249.77 ± 12.93 |
| Al | 1313.48 ± 286.71 | nd | nd | nd | nd | 523.28 ± 175.98 |
| Si | 17,613.75 ± 301.2 | 1813.69 ± 31.8 | 3754.1 ± 134.1 | 2848.18 ± 91.0 | 18,888.11 ± 244.4 | 31,076.83 ± 505.1 |
| K | 29,526.37 ± 433.3 | 11,776.67 ± 89.6 | 17,264.06 ± 254.5 | 25,069.97 ± 507.4 | 19,498.66 ± 165.0 | 26,804.84 ± 133.1 |
| P | 1137.55 ± 152.6 | 1220.13 ± 86.3 | 2243.13 ± 169.3 | 2344.26 ± 140.1 | 2431.09 ± 113.2 | 3542.42 ± 118.0 |
| Ca | 87,665.85 ± 1051.0 | 36,475.24 ± 147.4 | 40,163.07 ± 244.3 | 45,371.12 ± 710.7 | 20,635.21 ± 69.7 | 37,377.25 ± 267.3 |
| S | 4341.74 ± 75.33 | 1587.02 ± 15.64 | 3690.75 ± 39.90 | 3119.35 ± 49.08 | 3504.57 ± 60.99 | 4361.53 ± 54.27 |
| Cl | 5863.12 ± 53.14 | 371.34 ± 2.22 | 1221.23 ± 21.88 | 378.5 ± 20.99 | 3308.04 ± 30.19 | 5680.75 ± 88.63 |
| Fe | 9483.38 ± 47.55 | nd | 266.75 ± 30.33 | 91.57 ± 22.96 | 537.45 ± 21.29 | 1717.2 ± 46.76 |
| Zn | 92.02 ± 3.33 | nd | 22.78 ± 1.08 | 19.2 ± 0.39 | 14.17 ± 0.49 | 35.94 ± 4.65 |
| Ag | 6.61 ± 0.46 | 5.08 ± 0.06 | 6.03 ± 0.08 | 5.39 ± 1.28 | 4.85 ± 1.47 | 5.77 ± 1.05 |

Note: Data are expressed as means ± standard deviation; nd–not detected.

Another significant result from the analyses was the mineral composition of the samples, with some differences in specific elements, such as Fe, P, K, Ca, Cl, and S. The dif-

ferences in the metal content, which was twice as high in LW as the other samples and almost as high in M2, is another reason for the higher ash content [44].

Raveendran et al. [60] concluded that the mineral contents of biomass, in combination with the organic composition, play a major role in determining their properties as a fuel.

Boiler combustion temperature is also important to minimize both ash and emissions and to increase the energy value of the fuel. According to Wang et al. [61], significant weight loss occurs at temperatures of around 550 °C, and inorganic compounds decompose at temperatures above 750 °C. In this regard, it is necessary to remember that ISO 18122 [33] defines a testing temperature of just 550 ± 10 °C, but when the material is burnt in boilers, the temperature is higher, so even LW could produce a lower amount of ash.

The S content of the branch samples investigated varied between 1587 and 3690 mg·kg$^{-1}$; the non-wood samples had higher levels, between 4341 and 4361 mg·kg$^{-1}$. For comparison, the average S content of the bark samples varied between 307 and 499 mg·kg$^{-1}$, and that of the straw samples varied between 745 and 788 mg·kg$^{-1}$. The Cl content of the bark samples was between 371 and 3308 mg·kg$^{-1}$; again the non-wood samples were higher, between 5680 and 5863 mg·kg$^{-1}$. Typical concentrations of Cl in bark were approximately 220 mg·kg$^{-1}$, and in straw they were between 1150 and 4900 mg·kg$^{-1}$ [27].

A typical attribute of waste materials is increased concentrations of Cl and S elements, which can result in problems related to emissions and boiler corrosion. Other elements cause problems only in specific types of material; therefore, producers of biofuels must react accordingly by mixing different input raw materials [10,20–27].

All the test samples, especially LW and M2, had naturally high levels of S and CL, which is in agreement with the results of other studies. Even so, the samples still met the limits set out in the standards.

### 3.1.6. Heavy Metal Content

The heavy metal contents (Cd, Pb, Zn, Cr, Cu, As, and Hg) were determined and compared to the limits contained within the standards to ensure the protection of the environment.

In general, the heavy metal content is higher in the bark and leaves than in the wood and is related to natural accumulation. However, all samples met the heavy metal standard, except for Cd, which was over 20 times higher than the limit. In this context, it is noticeable that materials from the communal area were affected by emissions from tyre wear [62], and it is necessary to take this into consideration when planning their use [10].

### 3.2. Pellet Production Process

Operational tests were performed to check the feasibility of producing biomass pellets from green maintenance waste biomass. To determine the quality of these pellets, several characteristics were measured: diameter, length, moisture, and mechanical durability (Table 3).

**Table 3.** Average parameters of pellets after the pelletization process.

| ID | LW | BW | BB | BC | M1 | M2 |
|---|---|---|---|---|---|---|
| Sample | Walnut Leaves | Walnut Branches | Black Locust Branches | Common Apricot Branches | Mixture 1 Branches | Mixture 2 Perennials |
| Moisture after pelletization (%) | 11.54 ± 0.73 [c] | 8.34 ± 0.37 [ab] | 7.74 ± 0.2 [a] | 7.53 ± 0.5 [a] | 9.55 ± 0.16 [b] | 11.13 ± 0.68 [c] |
| Mechanical durability (%) | 96.26 ± 0.37 [b] | 98.64 ± 0.09 [a] | 98.80 ± 0.14 [a] | 98.49 ± 0.14 [a] | 98.02 ± 0.29 [a] | 96.99 ± 0.48 [b] |
| Diameter (mm) | 6.16 ± 0.2 [b] | 6.57 ± 0.18 [a] | 6.55 ± 0.21 [a] | 6.68 ± 0.22 [a] | 6.61 ± 0.15 [a] | 6.13 ± 0.17 [b] |
| Length (mm) | 10.02 ± 2.87 [d] | 17.58 ± 2.7 [ab] | 17.01 ± 3.12 [ab] | 15.64 ± 3.42 [a] | 20.29 ± 2.12 [c] | 19.26 ± 1.77 [bc] |

Note: The data are expressed as mean values ± the standard deviations. Means values accompanied by a letter in the row are significantly different, according to Tukey's test ($p \leq 0.05$).

### 3.2.1. Moisture and Pelletization

The moisture content of the input material is one of the most crucial factors that affects the pelletization process in terms of its efficiency, stability, and quality of the pellets. Raw materials with a high moisture content have greater drying requirements, and this has a negative effect on the costs of pellet production. Pellets with a high moisture content deteriorate due to microbial decomposition [50,63].

According to ISO 17225-2, graded wood pellets [53], for all quality classes, have a moisture limit of ≤10%, and according to ISO 17225-6, graded non-woody pellets [54] have a moisture limit of ≤12% (class A) and ≤15% (class B). According to the results obtained, the pellet moisture limit did not exceed in any of the samples. The values of the mean moisture content of the studied pellet samples ranged from 7.53 to 11.54%. The LW and M2 samples had the highest moisture content. For the LW sample, the higher moisture level resulted in the sample with the worst overall properties of all the input materials, the worst energy results, and the worst performance in the pelletization process, during which it was more difficult to maintain optimal pellet production, and the homogeneity of the pellets produced was worse than that of any other sample material. On the contrary, there were no real issues during the pelletization of the branch materials. It was not necessary to adjust the process settings during pelletization to match the flow of material; the pellets formed well, and all of their other resultant properties were satisfactory.

### 3.2.2. Mechanical Durability

During storage, transportation, and handling, pellets are mechanically damaged when they rub against each other, the container, or the conveyor walls. Grinding of the edges and surfaces of the pellets creates fine particles that are undesirable: it wastes material and creates dust, and there is a risk of ignition [64,65].

Bassam [58] stated that the durability of pellets can be improved, for example, by using sieves after the pelletizing process when the pellets are deliberately abraded during cooling. Gilvari et al. [66] state that storage had a minimal impact on the durability of pellets.

ISO 17225-2 [53] defines the mechanical durability of wood pellets as ≥97.5% (classes A1 and A2) and ≥96.5% (class B) for industrial use. ISO 17225-6 [54] for non-wood pellets defines mechanical durability as ≥97.5% (class A) and ≥96.0% (class B).

The mechanical durability of pellet samples BW, BB, BC, and M1 were classified as A1. The LW and M2 samples exhibited lower levels of mechanical durability and were classified as B.

Wood pellets have a higher mechanical resistance in comparison to pellets from other raw materials. The M2 sample had a higher level of mechanical durability than LW, which was likely primarily caused by the lower potential for the pelletization of LW materials. It was also likely affected by the higher content of essential oils, which bind the mixture, or the lower inorganic compound level, which leads to the poor binding of the mixture [63].

The results from Gilvari et al. [66], that mechanical resistance depends on the length of the pellets, were not confirmed, particularly due to the differences in individual raw materials.

### 3.2.3. Pellets Dimensions

The ISO 17225-2 standard [53] for wood pellets defines the length of a pellet for all classes and uses, as 3.15 to 40 mm with a diameter of 6 to 12 mm. ISO 17225-6 [54] for non-wood pellets defines the length for both classes as 3.15 to 40 mm with a diameter of 6 to 10 mm and 3.15 to 50 mm with a diameter of 12 to 25 mm. The diameter of the pellets is defined by the pelletization matrix, and the length can be modified (shortened) on the pelletizing machine, for example, with a knife. Good quality pellets should not crumble or swell significantly. These two parameters are not significant limitations from the perspective of their use: boilers and screw conveyors can handle this high degree of variability [64].

In terms of dimensional parameters, the pellets had a diameter greater than 6 mm due to material stress relaxation. The length of the pellets was 10–20.3 mm (Table 3). In

particular, the materials whose input properties were easy to modify and whose calorific values, together with other monitored parameters, were sufficiently high, had the potential to be used as suitable biofuels. In our case, we can recommend the branch samples (BW; BB; BC) and the perennial mix (M2).

## 4. Conclusions

In this study, typical raw materials from green area maintenance were harvested and analysed. They were used to produce biofuel pellets and were evaluated to establish their degree of compliance with industrial standards. The most important parameters (humidity, calorific value, and durability) met the requirements of the standards, especially the lower limit of the lower calorific value (14.9–16.2 MJ·kg$^{-1}$). However, the ash content (up to 9.5%), the moisture at collection (up to 16%), and the overall pelletizing process were not optimal. These results do not represent an obstacle to their use per se; it is only necessary to consider them when the mix is pelletized or burnt in a boiler. These raw materials can be mixed with other common materials, such as wood.

Of the selected materials, only fallen leaves were unsuitable for direct use, and it is recommended that their use be restricted to thermochemical processes, such as gasification, liquefaction, pyrolysis, or torrefaction.

These results are timely and helpful in identifying diverse materials suitable for the production of solid biofuel pellets, with or without reduced levels of wooden material. These results align with those of other studies, and the experience gained can be useful for biofuel producers or within the framework of urban waste management. From this perspective, it is necessary to be mindful of the costs of collecting and treating these materials (especially drying) and the economic viability of their use. Further research should focus on establishing the production potential of materials under urban conditions, collection technology, and utilization in decentralized power plants.

**Author Contributions:** Conceptualization, V.M. and P.B.; methodology, V.M.; software, V.M.; validation, J.S. and P.B.; formal analysis, V.S.; investigation, V.M. and L.V.; resources, P.B.; data curation, V.S. and L.V.; writing—original draft preparation, V.M.; writing—review and editing, P.B.; visualization, J.S.; supervision, V.S.; project administration, J.S.; funding acquisition, J.S. All authors have read and agreed to the published version of the manuscript.

**Funding:** This research was funded by the National Agency for Agricultural Research of the Ministry of Agriculture of the Czech Republic, grant number QK21020121, and the project of long-time development of the Research Institute of Agricultural Engineering, p.r.i. no. RO0623.

**Institutional Review Board Statement:** Not applicable.

**Informed Consent Statement:** Not applicable.

**Data Availability Statement:** Data are contained within the article.

**Conflicts of Interest:** The authors declare no conflict of interest.

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
