# Peer review of "Energy Potential of Urban Green Waste and the Possibility of Its Pelletization"

_sustainability, doi:10.3390/su152316489_

Round 1
Reviewer 1 Report
Comments and Suggestions for Authors
The manuscript entitled “Energy potential of urban green waste and the possibility of its pelletization” is of interest, however, it requires some revision.
General suggestions
Please, revise English to make some sentences more easy to read.
Abstract
The conclusive part of the abstract is not informative. It would be more useful to specify “…particularly problematic regarding a number of parameters”.
Introduction
Revise the sentence ” Such waste can in European cities that have a high level of green maintenance comprise 10 to 20% of the municipal solid waste and is available throughout the year with almost the same quality and moisture, particularly the solid and dry waste (trimmed wood of trees and bushes, dry plant matter, fallen leaves, etc.) from public green spaces and gardens [7,8].”. Make it more simple.
The introduction is mainly focused on the energy needs of the EU. However, there is lack of information is the proposed green waste suitable for the pelletization, if its amount is enough to make the approach cost-effective, etc. It would be better to add more supportive materials if there was a strong reason to conduct the proposed work.
Avoid duplication: “The aim of this study is to analyse the energetic characteristics of waste products 74 arising from the maintenance of park and communal greenery to characterise their suitability…”
Materials and Methods
The subsection 2.1. Climatic conditions
Is suggested to be transferred to the Results and discussion section as it doesn’t describe the experimental set up.
The link “The weather pattern in 2022 is available here: http://a.la-a.la/chart/cl.php?probe=11359248.” Should be presented as a reference at the end of the manuscript.
The background on which the sampling was conducted can also be transferred to the Results section: “Materials were selected that had not yet been verified in a similar context and had minimal common practice use. The materials are, however, produced in relatively large quantities during urban green maintenance.”
Figure 2 - Species names should be italicized.
Results and discussion
The discussion part of the section lacks the information of the practical value of the proposed research.
It mainly involves the information from the standards, however, a few information was observed on the similar research.
Conclusions
The section is too long. It partially duplicates the results.
Please, revise the section in order to highlight the key achievements and points of your work.
References
Revise some references to remove the uppercase font in some of them.

Comments on the Quality of English LanguagePlease, revise English to make some sentences more easy to read.
Author Response
Dear Reviewer,
I’m grateful for your comments and found them very helpful. I tried to incorporated most of them in order to imporve the article.
General suggestions
Please, revise English to make some sentences more easy to read.
Answer: We cooperated with professional English editors on wording, so I do hope we have a better text now.
Abstract
The conclusive part of the abstract is not informative. It would be more useful to specify “…particularly problematic regarding a number of parameters”.
Answer: Revised
Introduction
Revise the sentence ” Such waste can in European cities that have a high level of green maintenance comprise 10 to 20% of the municipal solid waste and is available throughout the year with almost the same quality and moisture, particularly the solid and dry waste (trimmed wood of trees and bushes, dry plant matter, fallen leaves, etc.) from public green spaces and gardens [7,8].”. Make it more simple.
Answer: Revised
The introduction is mainly focused on the energy needs of the EU. However, there is lack of information is the proposed green waste suitable for the pelletization, if its amount is enough to make the approach cost-effective, etc. It would be better to add more supportive materials if there was a strong reason to conduct the proposed work.
Answer: We haven´t evaluated the cost-effective site. Based on your recommendation, we added references and reflection on this idea as well.
Avoid duplication: “The aim of this study is to analyse the energetic characteristics of waste products 74 arising from the maintenance of park and communal greenery to characterise their suitability…”
Answer: Revised
Materials and Methods
The subsection 2.1. Climatic conditions
Is suggested to be transferred to the Results and discussion section as it doesn’t describe the experimental set up.
Answer: The journal recommends in its guidelines for authors to state general information on experiments in Materials and Methods.
The link “The weather pattern in 2022 is available here: http://a.la-a.la/chart/cl.php?probe=11359248.” Should be presented as a reference at the end of the manuscript.
Answer: Corrected
The background on which the sampling was conducted can also be transferred to the Results section: “Materials were selected that had not yet been verified in a similar context and had minimal common practice use. The materials are, however, produced in relatively large quantities during urban green maintenance.”
Answer: The journal recommends in its guidelines for authors to state general information on experiments in Materials and Methods.
Figure 2 - Species names should be italicized.
Answer: Corrected
Results and discussion
The discussion part of the section lacks the information of the practical value of the proposed research.
It mainly involves the information from the standards, however, a few information was observed on the similar research.
Answer: We have added citations, comparisons and practical information.
Conclusions
The section is too long. It partially duplicates the results.
Please, revise the section in order to highlight the key achievements and points of your work.
Answer: Revised
References
Revise some references to remove the uppercase font in some of them.
Answer: Corrected
Reviewer 2 Report
Comments and Suggestions for Authors
This paper introduces the energy potential of urban green waste and evaluates with industrial standards. It also delves into the characteristics of the pelletization process. The experimental findings offer valuable insights for the development of urban waste management frameworks and the exploration of opportunities for enhancing public green environments. However,some adjustments are required for specific points:
1.Please consider add one paragraph at the end of the introduction that outlines the structure of the research in the subsequent sections.
2.Please consider improving the introduction and related work sections by incorporating more references from the years 2022 or 2023.
3.Please provide a clearer explanation of what 'a' and 'ab' represent in Table 1 and Table 3.
4.Please estimating the power gain based on the power consumption of pellets and the energy values obtained from heating.
Comments on the Quality of English Language
Minor editing of English language required.
Author Response
Dear Reviewer,
we are grateful for your comments and found them very helpful. We tried to incorporated most of them in order to imporve the article.
1.Please consider add one paragraph at the end of the introduction that outlines the structure of the research in the subsequent sections.
Corrected and added
2.Please consider improving the introduction and related work sections by incorporating more references from the years 2022 or 2023.
Corrected and added new citations.
3.Please provide a clearer explanation of what 'a' and 'ab' represent in Table 1 and Table 3.
Corrected
4.Please estimating the power gain based on the power consumption of pellets and the energy values obtained from heating.
Added in the text
Reviewer 3 Report
Comments and Suggestions for Authors
Sustainability-2710459 presents a study on the Energy potential of urban green waste and the possibility of its pelletization. Six typical examples of input raw material were selected, analysed, and subsequently pelletized. The results of the experiment reflect a number of important findings. The paper contents are adequate to the scope of Sustainability. There are, however, several factors that prevent the publication of this article in its present form.
1. Most data in this paper are presented in the table, and in fact, would have worked better if it had been accompanied by a graph, such as a plot.
2. The main contribution of this paper could be added at the end of Chapter 1.
3. There seems to be a problem with the chapter numbers, chapter 3 followed by chapter 5?
4. There are some problems with the reference formatting of the article that need to be revised.
Comments on the Quality of English LanguageNone.
Author Response
Dear Reviewer,
we are grateful for your comments and found them very helpful. We tried to incorporated most of them in order to imporve the article.
1) Most data in this paper are presented in the table, and in fact, would have worked better if it had been accompanied by a graph, such as a plot.
We already tried to present findings in a graph, but the values of the parameters are too different to be combined into one graph (e.g. ashiness). Thus, the small differences between the individual variants would not be visible and the graph would lose its clarity. Furthermore, we work with too many parameters, so it would be difficult to put all into graphs. Therefore, we decide to rather use tabular form.
2) The main contribution of this paper could be added at the end of Chapter 1.
Added in the text
3) There seems to be a problem with the chapter numbers, chapter 3 followed by chapter 5?
Corrected
4) There are some problems with the reference formatting of the article that need to be revised.
Corrected
Round 2
Reviewer 1 Report
Comments and Suggestions for Authors
The manuscript was improved and can be accepted.